# The Role of NOX4 in Parkinson’s Disease with Dementia

**DOI:** 10.3390/ijms20030696

**Published:** 2019-02-06

**Authors:** Dong-Hee Choi, In-Ae Choi, Cheol Soon Lee, Ji Hee Yun, Jongmin Lee

**Affiliations:** 1Center for Neuroscience Research, Institute of Biomedical Science and Technology, Konkuk University, Seoul 05029, Korea; adia86@naver.com (I.-A.C.); slaoa1428@naver.com (C.S.L.); bsea78@empas.com (J.H.Y.); 2Department of Medical Science Konkuk University School of Medicine, Konkuk University, Seoul 05029, Korea; 3Department of Rehabilitation Medicine, Konkuk University School of Medicine, Konkuk University, Seoul 05029, Korea

**Keywords:** Parkinson’s disease, cognitive impairment, hippocampus, alpha-synuclein, amyloid beta, A11 oligomer, NADPH oxidase

## Abstract

The neuropathology of Parkinson’s disease with dementia (PDD) has been reported to involve heterogeneous and various disease mechanisms. Alpha-synuclein (α-syn) and amyloid beta (Aβ) pathology are associated with the cognitive status of PDD, and NADPH oxidase (NOX) is known to affect a variety of cognitive functions. We investigated the effects of NOX on cognitive impairment and on α-syn and Aβ expression and aggregation in PDD. In the 6-hydroxydopamine (6-OHDA)-injected mouse model, cognitive and motor function, and the levels of α-syn, Aβ, and oligomer A11 after inhibition of NOX4 expression in the hippocampal dentate gyrus (DG) were measured by the Morris water maze, novel object recognition, rotation, and rotarod tests, as well as immunoblotting and immunohistochemistry. After 6-OHDA administration, the death of nigrostriatal dopamine neurons and the expression of α-syn and NOX1 in the substantia nigra were increased, and phosphorylated α-syn, Aβ, oligomer A11, and NOX4 were upregulated in the hippocampus. 6-OHDA dose-dependent cognitive impairment was observed, and the increased cognitive impairment, Aβ expression, and oligomer A11 production in 6-OHDA-treated mice were suppressed by NOX4 knockdown in the hippocampal DG. Our results suggest that increased expression of NOX4 in the hippocampal DG in the 6-OHDA-treated mouse induces Aβ expression and oligomer A11 production, thereby reducing cognitive function.

## 1. Introduction

Parkinson’s disease (PD) is a common progressive neurodegenerative disease. The main neuropathological feature of PD is the loss of dopaminergic neurons [1]. Patients with PD have a high cognitive impairment and the disease has a serious negative impact on quality of life. It is now proven that 20%–40% of patients with PD will have a cognitive deficiency early in their illness. In addition, the risk of dementia in these patients is six times higher than it is in age-matched controls, and 80% of patients are expected to develop dementia after 20 years [2]. Parkinson’s neuropathology with dementia (PDD) is heterogeneous and involves several disease mechanisms [3]. Αlpha-synuclein (α-syn) pathology in the brainstem and cerebral cortex seems to exhibit the most potent neuropathological correlation with the dementia that occurs in PD [4]. Despite the central role of α-syn pathology in PDD, some studies have found that Aβ plaques and tau neurofibrillary tangles, a characteristic pathological cause of Alzheimer disease (AD), are related to the cognitive status of PDD [4,5,6,7,8,9]. Neuropathological examinations using a recently developed molecular imaging approach showed that the production of amyloid plaques is common in PDD with Lewy body dementia [5,10,11,12,13,14]. In 50% of patients with PDD, Aβ plaques and tau-containing nerve fibers are entangled, which can aggravate the prognosis of disease via synergism with α-syn pathology [4].

Emerging data on cognitive impairment in PD support the role of the hippocampus in cognitive function [15,16]. The hippocampus, which is a structure that exists in the temporal lobe, is involved in learning and memory processes [17,18]. Hippocampal dysfunction may account for the high prevalence of dementia at the stage of progression of PD [1]. Neuropathological studies have shown a link between the severity of cognitive impairment in PD and the degree of deposition of Lewy bodies and Lewy neurites in the hippocampus, suggesting an important role for hippocampal structures in cognitive impairment [1,19]. Imaging studies that examined the association between cognitive impairment in PD and hippocampal abnormalities [20,21,22] have revealed that the hippocampus affects PDD induction [20,21,22].

Recently, we reported that cognitive dysfunction in a rat model of vascular dementia is associated with an increase in the hippocampal neuronal NADPH oxidase (NOX) 1 [23]. New evidence indicates that NOX oxidases are involved in the cognitive dysfunction observed in various diseases [24]. Several important findings show that increased NOXs are observed in postmortem brain tissues of patients with AD and mild cognitive impairment (MCI) [25,26]. Studies of postmortem brain tissues of AD patients demonstrated an upregulation of the p67phox, p47phox, and p40phox isoforms and an increase in NOX enzymatic activity [25]. In another study of patients with MCI, NOX activity was increased in the temporal gyrus compared with that in the control group, and gp91phox and p47phox were upregulated in microglia and neurons [26]. These studies suggest that increased NOX-related redox pathways may contribute to AD progression [24,25,26]. The role of NOX in the cognitive dysfunction of PDD has not been established. Therefore, we investigated the effect of NOX on neuropathies such as α-syn and Aβ formation related with cognitive impairment of PDD.

## 2. Results

### 2.1. Motor Deficit, Loss of TH^+^ Cells, Increased Expression of α-Synuclein and NOX1 in the 6-OHDA-Lesion Side of the Striatum and Substantia Nigra

To evaluate motor function, we performed a rotation test with apomorphine (APO). APO-induced rotations toward the undamaged side were induced in 6-OHDA-treated mice (Figure 1a). Quantitative data showed a significant difference in the number of rotations between the 6-OHDA-injured mice and the control mice at 4 weeks after 6-OHDA injection. The number of rotations was increased, depending on 6-OHDA doses (F(3,44) = 8.881, *p* < 0.01; *p* < 0.001, Figure 1a). Unilateral medial forebrain bundle (MFB) lesions resulted in significant losses of tyrosine hydroxylase (TH) immunoreactive dopaminergic neurons in the striatum and substantia nigra (SN) in the 6-OHDA-lesion side (Figure 1b). We reported previously an increase in the expression of α-synuclein (α-syn) and NOX1 in the SN of mice or rats treated with paraquat or 6-OHDA [27,28,29]. In the 6-OHDA-injected mouse model, the expression of α-syn and NOXs was examined. We confirmed that the expression of α-syn was increased in 6-OHDA-treated mice. Similarly, the expression of NOX1 was increased after treatment with 6-OHDA, while the expression of NOX4 was not changed (Figure 1c,d).

### 2.2. Cognitive Deficit Induced by MFB Injury Caused by 6-OHDA Injection

The cognitive deficit of mice with unilateral MFB lesions was detected using the Morris water maze task. The latency of locating the platform and search errors were significantly increased in the spatial learning trials in a 6-OHDA dose-dependent manner (two-way repeated-measures analysis of variance (ANOVA), F(3,44) = 12.46, *p* < 0.05; F(3,44) = 17.53, *p* < 0.01, Figure 2a,b). There was no significant difference in the swimming speed (F(3.44) = 1.860, *p* = 0.1689, Figure 2c). There was a difference between the groups regarding the percentage of time spent in the target quadrant and in platform crossing number in the probe trials of 30 s (F(3,44) = 5.683, *p* < 0.01, Figure 2d,e). The percentage of time spent in the target quadrant and the number of target site crossovers were notably decreased in the probe trials in rats injected with 3 μg/kg of 6-OHDA (F(3,44) = 6.303, *p* < 0.05, Figure 2d,a,e) compared with those in control mice.

### 2.3. Levels of Alpha-Synuclein and Amyloid Beta Pathology in Animals with MFB Injury Caused by 6-OHDA Injection

Previous studies have shown that PDD has an increased α-syn pathology in the cerebral cortex, limbic area, parahippocampal gyrus, or anterior cingulate gyrus compared with PD [4,7,30,31]. Moreover, amyloid beta (Aβ) pathology in PDD is correlated with cognitive impairment [4].

Therefore, the expression of α-syn, phosphorylated α-syn (p-α-syn), Aβ, and A11 oligomers was confirmed in the cortex, striatum, SN, and hippocampus of 6-OHDA-treated mice by dot blot analysis. α-Syn expression was increased in the striatum and SN (*p* < 0.01, *n* = 6, Figure 3a,f), and p-α-syn was significantly upregulated in the SN and hippocampus (*p* < 0.05; *p* < 0.01, *n* = 6, Figure 3b,f). Specifically, the expression of Aβ was markedly increased in the hippocampus (*p* < 0.01, *n* = 6, Figure 3c,h), and the A11 oligomer was significantly increased in SN and hippocampus (*p* < 0.05; *p* < 0.01, *n* = 6, Figure 3d,g). Therefore, new evidence suggests that p-α-syn, Aβ, and A11 oligomers are upregulated in the hippocampus of the PDD animal model (Figure 3). 

### 2.4. Increased Levels of Phosphorylated Alpha-Synuclein, Aggregated Amyloid Beta, and NADPH Oxidase-4 in the Hippocampus

Hippocampal atrophy is associated with MCI in PD [15]. The hippocampus is an important brain structure that is related to learning and memory [32,33]. We attempted to identify molecular changes in the hippocampal region that occur during cognitive impairment in the model of PD. The expression of α-syn and NOXs (Nox1, Nox2, and Nox4) in the hippocampus was detected by western blot analysis (Figure 4a,b). P-α-syn and NOX4 were increased in total lysates of the hippocampal tissues of mice injected with 6-OHDA (Figure 4a,b). The immunohistochemical analysis revealed that α-syn was expressed in the hilar (CA4) mossy fiber terminal fields originating from the dentate gyrus (DG) of the hippocampus (Figure 4c). In addition, α-syn immunostaining (red) was mostly colocalized with neuron-specific class III beta-tubulin (Tuj-1)-positive neurons, which were immunostained in green in the hilar (CA4) mossy fiber terminal fields (as identified by the yellow staining observed after merging the two images). P-α-syn expression was greatly increased in the DG granule cell layer of the hippocampus at 4 weeks after 6-OHDA operation compared with that observed in sham-operated animals (*p* < 0.01, *n* = 6, Figure 4d,e). Interestingly, Aβ expression was also greatly increased in the hilar (CA4) mossy fiber terminal fields, supragranular layer (SG), and outer and middle molecular layers of the fascia dentata (*p* < 0.001, *n* = 6, Figure 4f,h). Aβ immunostaining (red) was mostly colocalized with Tuj-1-positive neurons, which were immunostained in green in the hilar (CA4) mossy fiber terminal fields (as identified by the yellow staining observed after merging the two images). NOX4 (green) and oligomer A11 (red) levels were increased in the hilar (CA4) mossy fiber terminal fields and granule cell layer of the DG. These two proteins were mostly colocalized, as identified by the yellow staining observed after merging the two images (*p* < 0.01, *n* = 6, Figure 4g,h).

To investigate the effects of these molecules on cognitive deficits in PDD model with 6-OHDA, changes in NOX4, NOX1, Aβ expression and A11 oligomer production in hippocampus and behavioral changes of 6-OHDA injected mice were analyzed by Pearson correlation coefficient. There was a high correlation between cognitive impairment and A11 oligomer production, NOX4, and Aβ expression, except for NOX1 expression (Figure 5). 

### 2.5. Inhibition of NADPH Oxidase-4 in Hippocampal Neurons Reduced Cognitive Impairment in a PD Animal Model

To verify the role of the expression of NOX4 in the DG in the cognitive impairment observed in the PD animal model, selective inhibition of NOX4 was achieved via the stereotaxic delivery of an adeno-associated virus 2 (AAV2) containing an shRNA targeting NOX4 to the hippocampal DG (Figure 6a,b,c). These vectors expressed the enhanced green fluorescent protein (EGFP) separately, as an index of transduction efficiency (Figure 6a). EGFP-expressing cells were detected in the DG of mice injected with scrambled shRNA/AAV2 (Scb shRNA) or Nox4 shRNA/AAV2 (Nox4 shRNA) viral particles (Figure 6b,c). NOX4 expression was mostly suppressed after Nox4 shRNA viral particle injection, which implies the efficient knockdown of NOX4 expression (*p* < 0.001, *n* = 6, Figure 6b,c). The RT-PCR and western blot analyses performed after AAV2 injection demonstrated NOX4 knockdown efficiency in hippocampal DG (*p* < 0.001, *n* = 6, Figure 6d,e). NOX activities were also determined from 4 weeks post-6-OHDA administration using Scb shRNA or Nox4 shRNA viral particles. We observed that NOX activation was significantly decreased by Nox4 shRNA/AAV2 treatment at 4 weeks post-6-OHDA administration (F(3,20) = 3.405, *p* < 0.05, Figure 6f).

The role of NOX4 in motor performance after 4 weeks of 6-OHDA administration was evaluated by the APO-induced rotation (F(3,44) = 11.54, *p* < 0.01) and the rotarod tests (F(3,44) = 4.654, *p* < 0.05). There was no difference between Scb shRNA and Nox4 shRNA treatment in 6-OHDA-injected mice (Figure 7a,b).

Next, we used Morris water maze (MWM) tests to assess the effects of NOX4 on cognitive function recovery in the PD model, by assessing learning and memory retention and spatial memory. Sham-operated control mice that received injection of Scb shRNAs were able to quickly find the diving platform during training, whereas mice that received Scb shRNA injections and were treated with 6-OHDA showed poor improvement during training compared with control mice (Figure 7c). The reduction in spatial memory observed in Scb shRNA-injected 6-OHDA-treated mice was significantly restored by inhibition of NOX4 using Nox4 shRNA particles (two-way repeated-measures ANOVA, F(3,44) = 15.53, *p* < 0.05, Figure 7c). No differences in the swimming speed were observed between the four groups (F(3,44) = 1.245, *p* = 0.3051, Figure 7d). The time spent in the target quadrant and the number of platform crossings were not different between the groups in the first probe test, but the time spent in the target quadrant and the platform crossings in the second probe test differed significantly in the 6-OHDA group with inhibition of NOX4 (percentage of time spent in the target quadrant, F(3,44) = 25.04, *p* < 0.001; number of platform crossings, F(3,44) = 16.45, *p* < 0.001, Figure 7e,f). In addition, in the novel object recognition (NOR) test, the reduction in the NOR discrimination ratio observed in 6-OHDA-treated mice that received Scb shRNAs was reversed by the delivery of the NOX4 shRNA (F(3,44) = 5.806, *p* < 0.01, Figure 7g).

### 2.6. NOX4 Knockdown Reduces 6-OHDA-Mediated Amyloid Beta and A11 Oligomer Upregulation in the Hippocampus

To investigate the mechanism of inhibition of cognitive impairment caused by NOX4 knockdown, we examined the effects of NOX4 upregulation in the hippocampal DG of MFB-injured mice on the expression of α-syn and Aβ on protein aggregation in hippocampal tissues. The levels of α-syn, Aβ, and protein aggregation were investigated in the hippocampus of each group of animals by dot blot and immunohistochemical analyses.

We found that α-syn expression did not differ between groups and that the increase in the levels of p-α-syn observed in the 6-OHDA group was not present in the group with NOX4 inhibition (Figure 8a,b). Aβ expression and A11 oligomer production were significantly increased in the Scb shRNA plus 6-OHDA groups, and these increases were reversed by NOX4 inhibition (amyloid β, F(3,20) = 34.28, *p* < 0.001; A11, F(3,20) = 19.42, *p* < 0.001; Figure 8a,b). Immunohistochemical evaluations revealed a significant increase in the immunoreactivity of Aβ in the hippocampus of mice that received injection of 6-OHDA in the MFB; this increase was reversed in the group exposed to 6-OHDA in which NOX4 was knocked down (*p* < 0.001, *n = 6*, Figure 8c,d). The involvement of NOX4 on production of the A11 oligomers in the hippocampus of 6-OHDA treated mice was also evaluated; the significant increase in A11 immunoreactivity observed in the hippocampus of mice exposed to Scb shRNA plus 6-OHDA was reversed by NOX4 knockdown (Nox4 shRNA plus 6-OHDA) (*p* < 0.001, Figure 8e,f). Taken together, our results are highly suggestive of an active role for NOX4 in hippocampal Aβ pathology induced by 6-OHDA injection into the MFB, both at the transcriptional level and via a posttranslational aggregation mechanism.

## 3. Discussion

In this study, we observed cognitive impairment in a PD model in which 6-OHDA was administered to the MFB. We also demonstrated, for the first time, that this cognitive impairment was mediated by the expression and aggregation of Aβ in the hippocampus and was associated with an increase in NOX4 expression in the hippocampal DG.

Accumulation of p-α-syn, Aβ, and NOX4 was detected in the hippocampus after MFB injury via 6-OHDA injection. In addition, hippocampal Aβ expression, A11 oligomer formation, and cognitive impairment were attenuated by genetic interventions targeting NOX4 in the hippocampal DG. These results strongly indicate that NOX4-mediated hippocampal changes may serve as critical upstream processes in Aβ expression, A11 formation, and cognitive impairment in PDD.

We also found that dopaminergic neuronal death, a typical phenomenon in the SN and striatum of this PD model, and upregulation of NOX1 and α-syn were similar to the results reported in previous studies [27,28,29].

Cognitive impairment without functional deficits (as a diagnosis of dementia) is a common feature of PD and is predominant in the late stage of the disease [4,34,35]. The prevalence of cognitive impairment in PD is ~30%, and ~80% of patients with PD develop dementia during disease progression [4,34,36,37]. The incidence of progression to dementia in patients with PD is estimated to be more than four times that of the general population [4,36,38,39,40], which has a major impact on the quality of life of patients [4,36,38,39,40]. Therefore, it is important to understand the pathophysiology that causes cognitive dysfunction in PD, because cognitive impairment in this disease and the onset and progression of dementia are important for patient management and prognosis [4].

The MFB lesion model [41] used in this study was suitable for mimicking PDD because it showed cognitive impairment in the Morris water maze test depending on the concentration of the 6-OHDA that was injected into the MFB (Figure 2). In this PD model, the typical nigrostriatal dopaminergic degeneration [27,28,29] was observed (Figure 1). Lewy bodies and neuritic pathologies in the cerebral cortex are predominant in the neuropathology of PDD compared with PD, as confirmed by α-synuclein immunohistochemistry [3,4,5,30,31]. In many studies, PDD is distinguished from PD based on the global cortical and limbic α-syn expression [4,30,42,43]. Moreover, several studies have shown that the severity of α-syn accumulation differs in areas such as the parahippocampal gyrus or anterior cingulate gyrus [4,7,30]. However, even in patients with PD [4,30,31] or Lewy body dementia (LBD) [4,44,45], the cortical and limbic α-syn pathology is highly dependent on the individual, and α-syn pathology is inconsistent in patients with PDD [3,4,46]; therefore, other factors need to be studied for diagnostic accuracy. In this study, α-syn expression was confirmed in the cortex, striatum, SN, and hippocampus, thus confirming the α-syn pathology reported by existing studies. Increased α-syn expression was observed in the striatum and SN, and p-α-syn, which has a dominant α-syn activity, was significantly upregulated in the SN and hippocampus (Figure 3). In particular, the increase in the levels of p-α-syn in the hippocampus may pinpoint a new role for the hippocampus in the cognitive impairment observed in PDD.

Although α-syn pathology plays a key role in PDD, studies have shown that Aβ plaques and tau nerve fiber entanglement, which are characteristic pathological causes of AD, are associated with cognitive status in patients with PDD [4,5,6,7,8,9]. In the striatum of these patients, α-syn pathological lesions [4,43,47] and diffuse Aβ plaques (characteristic of AD) were found more frequently than they were in patients with PD without dementia [4,13]; this striatal pathology has been regarded as a factor that can cause cognitive impairment. Therefore, Aβ pathology, which is one of the neuropathologies of AD, may play an important role in the pathogenesis of PDD [4]. Based on this, we found that the expression of Aβ in the animal model used in the present study was slightly increased in the striatum and SN and dramatically increased in the hippocampus. The level of oligomer A11 also showed a significant increase in the SN and hippocampus. These results suggest that α-syn and Aβ pathology in the hippocampus affect the pathogenesis of PDD (Figure 3).

Studies have shown that oxidative damage caused by excessive reactive oxygen species (ROS) production affects the cognitive impairment associated with age, AD, vascular dementia (VaD), and mild cognitive impairment [23,24,25,26,48,49]. Nicotinamide adenosine dinucleotide phosphate oxidase (NOX) is one of the most important factors in the formation of ROS in the brain. Recently, we demonstrated that NOX1-mediated oxidative stress plays an important role in hippocampal neuronal degeneration and cognitive dysfunction in a vascular dementia animal model [23,24]. Several studies have shown that cognitive impairment in age- or AD-related transgenic mice occurs through an increase in NOX activity and an increase in the expression of NOX2 and NOX4, which lead to oxidative stress and amyloid deposition [24,48,50,51,52]. In addition, an increase in the NOX-related redox mechanism was observed in postmortem brain tissues of patients with AD, and it was suggested that these changes could lead to the progression of AD [25,26].

To investigate the role of NOX in the hippocampal α-syn and Aβ pathology involved in PDD, the expression of NOX in the hippocampus of 6-OHDA-treated mice was examined. We found that NOX4 was significantly upregulated in this brain structure, whereas NOX1 and NOX2 remained unchanged (Figure 4). Specifically, the expression of NOX4 dominated the DG region of the hippocampus and was colocalized with the expression of oligomer A11 (Figure 4). Changes in NOX4 expression were highly correlated with cognitive impairment (Figure 5). In mice carrying NOX 4 suppression through genetic manipulation (Figure 6), the cognitive decline that was detected after 6-OHDA administration was reduced (Figure 7) and the upregulation of Aβ and increased generation of A11 oligomer were attenuated (Figure 8). These results suggest that the cognitive impairment observed in PDD is related to the expression of Aβ and the production of A11 oligomer, which are induced by the increased activity of NOX4 in the hippocampus.

In the present study, we used a mouse model in which 6-OHDA was administered to the MFB to investigate the cause of cognitive impairment and progression to PDD in patients with PD. We showed that the upregulation of NOX4 in the DG of the hippocampus was accompanied by cognitive impairment and led to increases in Aβ expression and oligomer A11 production. In addition, it is necessary to overcome the limitations of the model by conducting studies using PDD models other than the 6-OHDA administration model, but our study may be useful for preclinical studies of new diagnostic and therapeutic technologies for PDD. In the future, it will be necessary to identify the mechanism underlying the involvement of NOX4 in Aβ synthesis and A11 production.

## 4. Materials and Methods 

### 4.1. Animals

A meta-analysis of recent global data has shown that the prevalence of PD is more frequent in men than in women aged 50–79 years [53,54]. A total of 96 male C57/BL6 mice ((12 weeks, weighing 27–30 g; Samtako BioKorea. Co. Ltd., Osan, Gyeonggi Korea; 24 sham control mice (sham control) and 72 6-OHDA injected mice, 6-OHDA, Sigma-Aldrich, Saint Louis, MO, USA)) were enrolled in this study and maintained in a room at 22 °C under a 12 h light/dark cycle; chow and water were provided ad libitum. The room was illuminated by incandescent lamps (luminous flux, 11.77 lumen). Animal treatments, including anesthesia and euthanasia, were carried out in accordance with the Principle of Laboratory Animal Care (NIH publication No. 85-23, revised 1985). All experimental procedures were approved by the Animal Experiment Review Board of Institutional Animal Care and Use Committee (IACUC) of Konkuk University (Permit Number: KU16191; permission date: 22 November 2016).

### 4.2. Animal Treatment and Unilateral 6-OHDA Lesion in MFB 

Under the ketamine (50 mg/kg) and xylazine (5 mg/kg) mixture through intraperitoneal (i.p.) injection, mice in the 6-OHDA group were fixed in a stereotaxic apparatus (Stoelting Co., Wood Dale, IL, USA), and then an incision was made on the scalp. The 6-OHDA solution (1, 2, or 3 μg/2 μL 6-OHDA dissolved in 0.9% saline containing 0.01% ascorbic acid as antioxidants) was injected unilaterally into the MFB using a 10 μL Hamilton microsyringe at one site of MFB using the following coordinates according to the Allen Mouse Brain Atlas [55,56,57]: 1.0 mm anterior to the bregma; 1.2 mm lateral to the midline; and 4.0 mm below the dura mater. Injections were carried out at a rate of 0.2 μL/min and a volume of 2 μL was injected in the site. The needle was left at the injected sites for 5 min before being withdrawn. Mice in the control group received the same volume of saline in the same injected sites (*n* = 12/group).

### 4.3. Apomorphine-Induced Rotation. 

The animals were injected subcutaneously with 0.5 mg/kg apomorphine hydrochloride (APO, Sigma-Aldrich, St. Louis, MO, USA). The effect of APO on motor asymmetry and the rotation to uninjured side (to the left in our study) in a 60 min period were recorded by two examiners that were blinded to animal states. All mice were measured at 4 weeks after 6-OHDA injection [57]. 

### 4.4. Immunohistochemistry

Following perfusion with saline and 4% paraformaldehyde in phosphate-buffered saline (PBS, GibcoBRL, Gaithersburg, MD, USA), brains were removed, and the forebrain and midbrain blocks were immersion-fixed in 4% paraformaldehyde and cryoprotected in sucrose. Serial coronal sections (30 μm) were cut on a cryostat, collected in cryopreservative, and stored at –20 °C. For immunolabeling studies, sections were incubated with a blocking solution (5% horse serum and 0.3% Triton X-100 in PBS, pH 7.5) and then with primary antibody at 4 °C overnight. Mouse monoclonal anti-TH antibody (MAB318, 1:2000) was obtained from EMD Millipore (Billerica, MA, USA). Next, sections were incubated with biotinylated goat anti-mouse IgG secondary antibodies (BA-9200,1:200, Vector Laboratories, Burlingame, CA, USA) in a blocking solution at room temperature for 1 h. The sections were incubated with avidin–biotin–peroxidase complex (Vector Laboratories, Burlingame, CA, USA) in PBS/Triton X-100 at room temperature for 1 h. Signal labeling was achieved with 0.05% 3,30-diaminobenzidine and 0.003% H_2_O_2_ (Vector Laboratories). Signals were evaluated on an inverted light microscope with 10× or 20× objectives (Carl Zeiss, Jena, Germany) [27].

### 4.5. Western Blot and Dot Blot Analysis

Tissues were washed with ice-cold PBS and lysed on ice in RIPA buffer (1% PBS, 1% NP-40, 0.5% sodium deoxycholate, 0.1% SDS) containing a protease inhibitor mixture (AEBSF, aprotinin, bestatin hydrochloride, E-64-[*N*-(trans-epoxysuccinyl)-l-leucine 4-guanidinobutylamide], leupeptin, pepstatin A) and phosphatase inhibitors (Sigma-Aldrich, St. Louis, MO, USA). A total of 30 μg of soluble protein per lane was loaded in SDS-PAGE and electrotransferred onto a PVDF membrane. Specific protein bands were detected by using specific anti-TH (Millipore, USA), α-synuclein (α -syn, BD Bioscience, Franklin Lakes, NJ, USA), p-α-syn (Abcam, Cambridge, UK), NOX1, NOX2, and NOX4 antibodies (Santa Cruz Biotechnology, Santa Cruz, CA, USA) and enhanced chemiluminescence (Pierce, Rockford, IL, USA) [29]. For dot blot, brain tissues were homogenized in a buffer containing 0.32 M sucrose, 1 mM NaHCO_3_, 1 mM MgCl_2_, 0.5 mM CaCl_2_, and 1% of a protease inhibitor mixture (AEBSF, pepstatinA, E-64, bestatin, leupeptin, and aprotinin). The soluble fraction was obtained by centrifugation at 1000× *g* and 5 μL of each sample, containing the same amount of protein, was spotted in a PVDF membrane. The membrane was air-dried for 4 h and blocked overnight at 4 °C in a 5% nonfat dry milk TBST (10 mM Tris-HCl, pH 7.8, 100 mM NaCl, 0.05% Tween 20) solution. Protein spots were detected using the primary antibodies (α-syn, BD Bioscience, USA; p-α-syn, Abcam, Cambridge, UK, Amyloid beta, Aβ, BioLegend (6E10), San Diego, CA, USA; A11 oligomer, Invitrogen, Waltham, MA USA) and secondary antibody conjugated to hydrogen peroxidase followed by enhanced chemiluminescence (ECL) detection [29].

### 4.6. Behavior Test

The MWM test was employed to evaluate the learning and memory of mice [23]. The MWM consisted of a large circular pool (150 cm in diameter, 45 cm in height), filled to a depth of 30 cm with water that was maintained at 28 ± 1 °C. The water was made opaque using a nontoxic white-colored dye. The tank was divided into four equal quadrants using two threads that had been fixed at right angles to each other on the rim of the pool. A submerged platform (15 cm in diameter) painted in white was placed inside the target quadrants of this pool, 1.5 cm below the surface of the water. The position of the platform was not changed throughout the training session. Four consecutive sessions, each consisting of five trials for 2 days (alternating two or three trials per day) was conducted over eight consecutive days. The hidden platform was always placed in the southeast quadrant of the pool. The mouse was gently placed in the water of the pool between quadrants, facing the wall of the pool, and the drop location was changed for each trial. The mice were then allowed 60 s to locate the submerged platform and were allowed to stay on the platform for another 30 s. If the mice failed to find the platform within 60 s, it was guided gently onto the platform and allowed to remain there for 30 s. The intertrial interval was 1 min. Performance accuracy was evaluated based on the analysis of the search error and the time latency data of all trials. A probe trial was conducted at 1 min after every 10^th^ training trial. The entire training procedure included two probe trials for each rat, during which the mice swam with the platform retracted to the bottom of the pool for 30 s. After recording the swimming path, the platform was raised to its normal position for completion of the trial. The time spent swimming in the target quadrant of the retracted platform was used as a parameter for the retention of spatial memory [23]. 

### 4.7. Double-Fluorescence Immunostaining of Tissues

Free-floating sections (30 μm) were incubated in 0.1 M PBS containing 5% normal donkey serum and 0.3% Triton X-100 for 1 h, and subsequently incubated overnight with specific primary antibodies (Tuj-1, 1:1000, BioLegend, San Diego, CA, USA; α-syn, 1:1000, BD Bioscience, USA; p-α-syn, 1:1000, Abcam, Cambridge, UK, Amyloid beta, Aβ, 1:1000, BioLegend (6E10), San Diego, CA, USA; A11, 1:1000, Invitrogen, Waltham, MA USA; NOX4 (1:500, Santa Cruz Biotechnology, USA) in 2% normal donkey serum (Vector Laboratories, Burlingame, CA, USA) in PBS at 4 and incubated with a 1:200 dilution of Alexa Fluor-conjugated donkey anti-rabbit (488) or donkey anti-mouse (546) antibodies (Invitrogen, Grand Island, NY, USA) for 1 h at room temperature and mounted on glass slides using Vectashield (Vector Laboratories, Burlingame, CA, USA). Fluorescent signals were evaluated on a confocal microscope (LSM 710, Carl Zeiss, Oberkochen, Germany) [29].

### 4.8. Quantitative Analysis

Sections including the hippocampus from six mice per group were subjected to analysis. Six regions of interests (ROIs) of 0.1 mm^2^ per one section in the hippocampus (1.5 ~ 2.5 mm posterior to bregma) were selected. The number of TH-, Tuj-1-, α-syn-, p-α-syn-, Aβ-, NOX4-, A11 oligomer-, or EGFP-positive cells was counted in each ROI and averaged. Data are represented as a percentage of total cell and signal-positive cell counts. All quantitative analyses were carried out in a blind manner.

### 4.9. Establishment of U6-NOX4 shRNA-CMV-EGFP/AAV and AAV Vector and AAV Viral Package.

The U6 promoter-driven shRNA expression system was established in the AAV2 vector. EGFP expression was separately controlled by a CMV promoter as a marker for the transduction efficiency. NOX4 shRNA was designed based on the siRNA sequence which efficiently knocked down NOX4 expression in HT22 cells. Mouse NOX4 shRNA sequence (5′-AGCTTAAGCAACATTTGGTGTCCACTTTAATTCAAGAGATTAAAGTGGACACCAAATGTTG CTTTTTTTTG-3′) was inserted between HindIII and EcoRI sites in the U6-CMV-EGFP/AAV vector. Both Scb shRNA and NOX4 shRNA/AAV were co-transfected with pHelper and pAAV-RC to HEK293 cells using a standard calcium phosphate method. After 72 h, the cells were harvested and crude recombinant AAV (rAAV) vector solutions were obtained by repeated freeze/thaw cycles. The cleared crude lysate was then applied on a heparin-agarose column (Sigma, St. Louis, MO). After all the lysate went through the column, the matrix was washed twice with 25 mL of PBS (pH 7.4, 0.1 M NaCl). The virus was then eluted with 15 mL of PBS (pH 7.4, 0.4M NaCl). The elutes was concentrated to about 1 ml with a Millipore Centriplus YM-30 Centrifugal Filter by centrifugation at 4000 rpm for 15–40 min. To adjust the NaCl concentration to physiological levels, the filter device was refilled with PBS (pH 7.4), and the virus was concentrated to 250–300 μL again. After removal of the virus-containing solution, the membrane of the filter device was washed three times with 100 μl of PBS (pH 7.4), which was added to the main part of the recombinant AAV2. The fractions containing high-titer rAAV vectors were collected and used for injection into animals. The number of rAAV genome copies was semiquantified by PCR within the CMV promoter region using primers 5′-GACGTCAATAATGACGTATG-3′ and 5′-GGTAATAGCGATGACTAATACG-3′. The final titers were 5.5 × 10^11^ genomes/mL (rAAV2-Scb shRNA), and 6.5 × 10^11^ genomes/mL (rAAV2-NOX4 shRNA) [27].

### 4.10. Adeno-Associated Virus 2-Mediated NOX4 Knockdown

AAV particles containing either NOX4 shRNA/AAV or Scb shRNA/AAV were stereotaxically injected into the hippocampal DG for 3 weeks before 6-OHDA or sham operation (*n* = 12/group). Mice were deeply anesthetized (ketamine and xylazine mixture 30 mg/kg, intraperitoneal) and placed in a mouse stereotaxic apparatus. NOX4 shRNA/AAV and Scb shRNA/AAV were then injected into both sites in the hippocampal DG (coordinate: anteroposterior, −2.0 mm; mediolateral, −1.5 mm; dorsoventral, −2.25 mm). A total of 1 × 10^11^ genome copy/mL recombinant AAV particles encoding NOX4 shRNA, or Scb shRNA diluted in 2 μL ice-cold sterilized PBS were used in every animal. The injection rate was 0.5 μL/min, and the syringe was kept in place for an additional 5 min before being retracted slowly. Mice were subjected to PBS or 6-OHDA injection after 3 weeks (*n* = 12 per group). The mice were allocated into four groups: sham-operated mice injected with Scb shRNA/AAV particles (Scb shRNA control, *n* = 12); sham-operated mice injected with Nox4 shRNA/AAV particles (Nox4 shRNA control, *n* = 12); 6-OHDA-injected mice with Scb shRNA/AAV particles (Scb shRNA 6-OHDA, *n* = 12); and 6-OHDA-injected mice with NOX4 shRNA/AAV particles (Nox4 shRNA 6-OHDA, *n* = 12). Twelve mice per group were used for the behavioral test, and 6 mice per group were used for the histological and biochemical studies.

### 4.11. NOX Activity Assay

NADPH oxidase activity was measured by a lucigenin-derived chemiluminescence assay as described [21,22,23]. Briefly, 5–7 mg homogenized protein was incubated with its substrate NADPH (100 mM) in a phosphate buffer (50 mM, pH 7.0) containing 150 mM NaCl and 1 mM EGTA for 15 min, followed by an addition of 5 mM lucigenin for 15 min in dark. The chemiluminescent signal (photon emission) was measured using a Turner 20/20 luminometer (Turner Designs, Sunnyvale, CA, USA). No activity could be measured in the absence of NADPH [58].

### 4.12. Novel Object Recognition Test

The mice were placed in a 40 × 40 × 40 cm (height)-sized box and allowed to equilibrate to the testing area 10 min per day for 3 days. After 24 h, two objects were placed in the testing area and then, the mouse was allowed to explore the two objects in the testing area for 5 min before being returned to the cage. After a 3 h interval, one of the objects was either relocated or replaced with a new object and the mice were allowed to explore the testing area once again for 2 min [23,59]. The intertrial interval between these two tests was 24 h. Exploring the novel object was defined if the center of the mouse’ head was oriented within 45° of the object and within 4 cm of it. Climbing over or sitting on an object was not included. A video camera was positioned over the arena, and exploratory behaviors were videotaped for later analysis. Exploratory time spent for novel objects was recorded, and the discrimination ratio was computed as [(time spent exploring novel object – time spent exploring familiar object)/total time spent exploring both objects] in a blind manner [23].

### 4.13. Data Analysis and Statistics

The MWM test was analyzed using a two-way repeated-measures ANOVA, followed by a post hoc least significant differences multiple comparisons test. The APO-induced rotation, swimming speed, probe test, NOR test, intensities of western blot, dot blot, and RT-PCR results, and cell counts after immunostaining were analyzed using a one-way ANOVA followed by a Newman–Keuls multiple comparisons test. If the value to be compared is a comparison between the two groups, a t-test was performed. Data are expressed as percentages of values obtained in control conditions and are presented as mean ± S.E.M. Null hypotheses of no differences were rejected if *p* < 0.05. All data analyses were performed using the SPSS version 22.0 software (IBM Corporation, New York, USA). The correlation coefficient was used to measure the strength of the relationship between the relative changes of the latency time in MWM and protein expression s in hippocampus (GraphPad, San Diego, CA, USA).

## Figures and Tables

**Figure 1 ijms-20-00696-f001:**
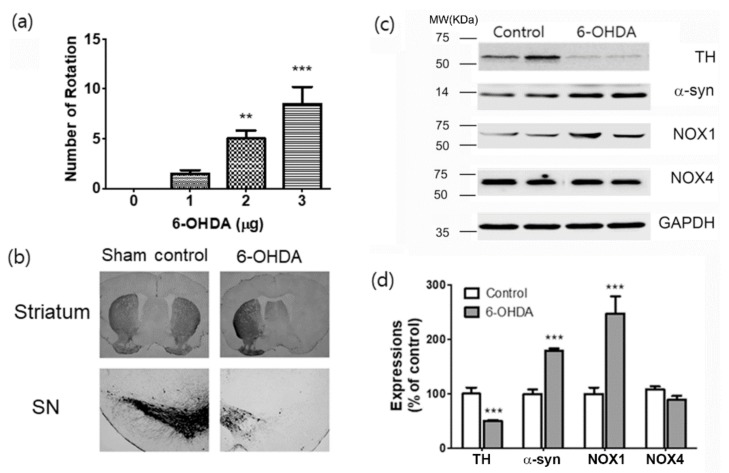
Evaluations of motor performance, dopaminergic neuronal death, expressions of α -synuclein and NOX1 in the 6-OHDA-induced PD mouse. (**a**) Total apomorphine (APO)-induced rotation numbers were counted at 4 weeks after 6-OHDA injection. (**b**) Representative photomicrographs of tyrosine hydroxylase (TH) staining in the mouse striatum and substantia nigra (SN) sections. (**c**) Results are presented as the mean ± SEM, *n* = 6. (**c**) Representative photomicrographs of western blots for α-synuclein (α-syn), NOX1, NOX4, and GAPDH in total lysates of the SN tissue at 4 w after 6-OHDA injection. MW, molecular weight; KDa, KiloDalton. (d) Signal intensities were measured using Quantity One software and are shown as a percentage of control. GAPDH was considered as an internal control. Results are presented as the mean ± SEM, *n* = 6. *** *p* < 0.001 vs. control.

**Figure 2 ijms-20-00696-f002:**
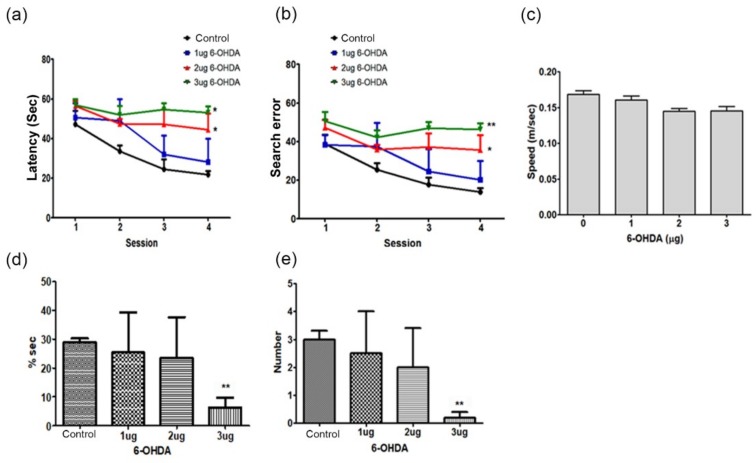
Cognitive impairment in the 6-OHDA-induced PD mouse. Spatial memory evaluation using the time latency (**a**), search error (**b**), swimming speed (**c**), percentage of time in target quadrant (**d**), and time latency in visible platform test (**e**). *n* = 12 per group, * *p* < 0.05, ** *p* < 0.01 vs. control.

**Figure 3 ijms-20-00696-f003:**
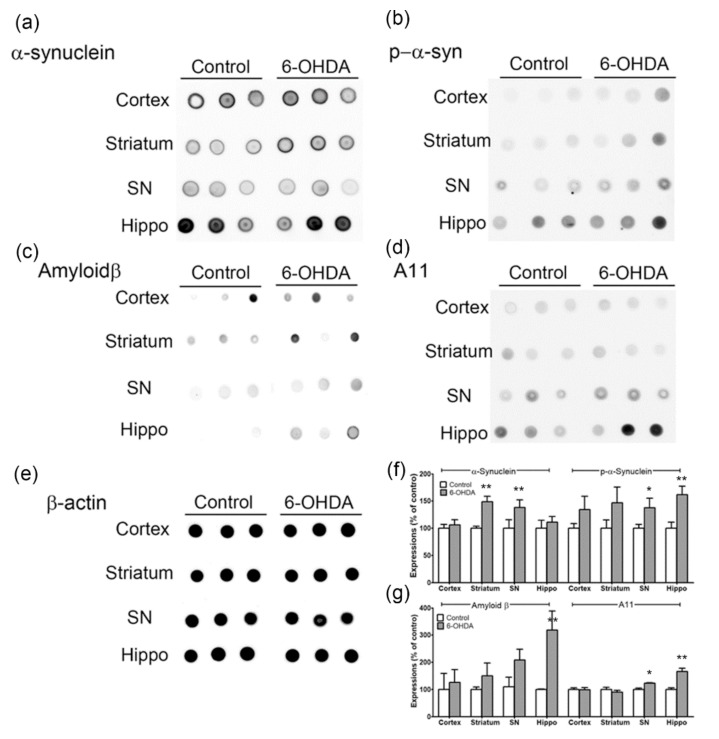
Expressions of α-synuclein, p-α-syn, amyloid β, A11 oligomers in the cortex, striatum, SN, and hippocampus of 6-OHDA-injectecd PD mice. Representative immunoblot analysis of (**a**) α- synuclein, (**b**) p-α-syn, (**c**) amyloid β (**d**) A11 oligomers, and (**e**) β-actin in cortex, striatum, SN, and hippocampus of 6-OHDA-injectecd PD mice. (**f** and **g**) All expression levels were quantified using Quantity One software and normalized against β-actin. The results are expressed as a percentage of control. Data are shown as the mean ± SEM. *n* = 6/group, * *p* < 0.05, ** *p* < 0.01, vs. control. p-α-syn, phosphorylated α-synuclein; SN, substantia nigra; Hipp, hippocampus.

**Figure 4 ijms-20-00696-f004:**
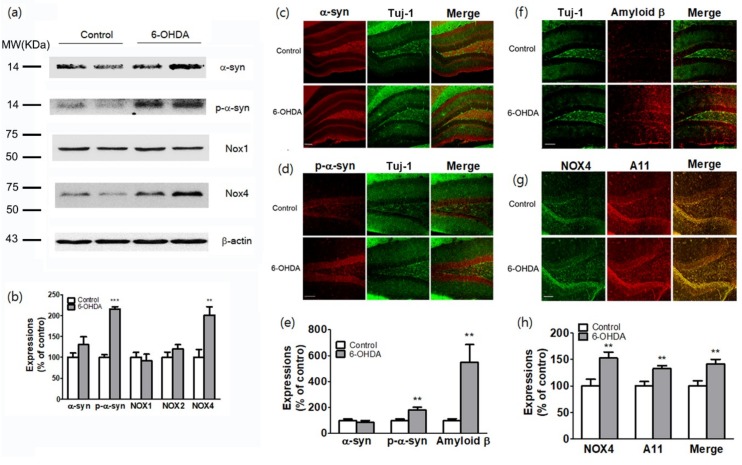
Expressions of p-α-syn, amyloid β, A11 oligomers, and NOX4 in the hippocampus of 6-OHDA-injectecd PD mice. (**a**) Representative western blot analysis of α-syn, p-α-syn, NOX1, NOX2, and NOX4 in the hippocampus of 6-OHDA-injectecd PD mice. MW, molecular weight; KDa, KiloDalton. (**b**) Expression levels were quantified using Quantity One software and normalized against β-actin. The results are expressed as a percentage of control. Data are shown as the mean ± SEM. *n* = 6 per group, ** *p* < 0.01; *** *p* < 0.001, vs. control. Representative photomicrographs of α-syn (**c**) and p-α-syn (**d**) staining in the mouse hippocampal sections. (**e**) Expression levels were quantified using Quantity One software. The results are expressed as a percentage of control. Data are shown as the mean ± SEM. *n* = 6. ** *p* < 0.01 vs. control. Representative photomicrographs of amyloid β (**f**), NOX4, A11, and NOX4 and A11 co-stating (**g**) in the mouse hippocampal sections. (**h**) Expression levels were quantified using Quantity One software. The results are expressed as a percentage of control. Data are shown as the mean ± SEM. *n* = 6. ** *p* < 0.01 vs. control. α-syn, α-synuclein; p-α-syn, phosphorylated α-synuclein; Tuj-1, neuron-specific class III beta-tubulin; scale bars: 50 μm.

**Figure 5 ijms-20-00696-f005:**
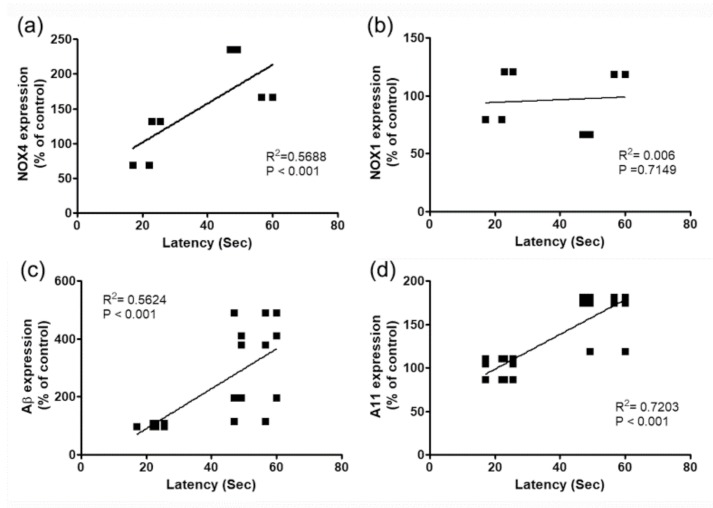
Relationship between the cognitive impairment and percentage of changes (%) of NOX4 (**a**), NOX1 (**b**), and Aβ expressions (**c**), and A11 oligomer production (**d**) in the hippocampus of 6-OHDA-injectecd PD mice.

**Figure 6 ijms-20-00696-f006:**
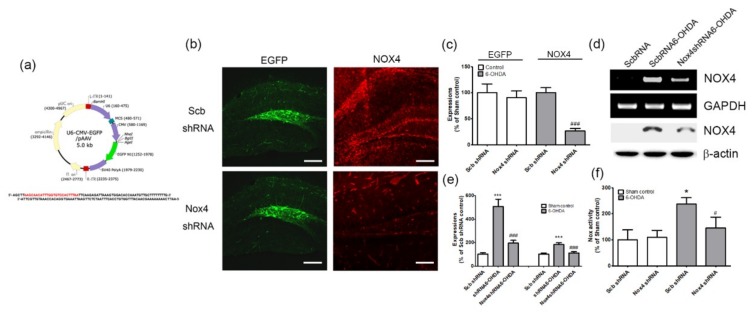
The transduction efficiency of scramble shRNA or NOX4 shRNA expression in hippocampus and reduced NOX4 expressions by NOX4 knockdown in 6-OHDA injected mice. (**a**) Establishment of U6 promoter-based NOX4 shRNA/AAV vector. The U6 promoter-driven shRNA expression system was established in the AAV vector. EGFP expression is separately controlled by a CMV promoter as a marker for the transduction efficiency. The NOX4 shRNA sequence was designed based on the siRNA sequence (boxed red nucleotides). (**b**) Representative photographs of tissue sections expressed EGFP (green) from mice hippocampal DG taken from mice injected with Scb shRNA/AAV2 or NOX4 shRNA/AAV2 and PBS (sham control) or 6-OHDA injection. To confirm NOX4 knockdown efficiency, NOX4 (red) expression was determined by NOX4 immunostaining in the hippocampal DG of 6-OHDA injected mice with shRNA/AAV2 or NOX4 shRNA/AAV2. (**c**) EGFP and NOX4 expression levels were quantified using Quantity One software. The results are expressed as a percentage of control. Data are shown as the mean ± SEM. *n* = 6. ### *p* < 0.001 vs. Scb shRNA 6-OHDA. (**d** and **e**) NOX4 knockdown efficiency in the hippocampal DG was also verified by both western blot analysis and RT-PCR performed after Scb shRNA or NOX4 shRNA injection. *n* = 6 per group. Expression levels were quantified using Quantity One software. The results are expressed as a percentage of control. Data are shown as the mean ± SEM. *** *p* < 0.001 vs. Scb shRNA control; ### *p* < 0.001 vs. Scb shRNA 6-OHDA. (f) NOX enzyme activity was measured in lysates of hippocampal tissue at 4 weeks after 6-OHDA treatment in Scb shRNA/AAV2 or NOX4 shRNA/AAV2-injected mice. *n* = 6 per group. * *p* < 0.01 vs. Scb shRNA control; # *p* < 0.05 vs. Scb shRNA 6-OHDA. Scb shRNA, scrambled shRNA; scale bars = 100 μm.

**Figure 7 ijms-20-00696-f007:**
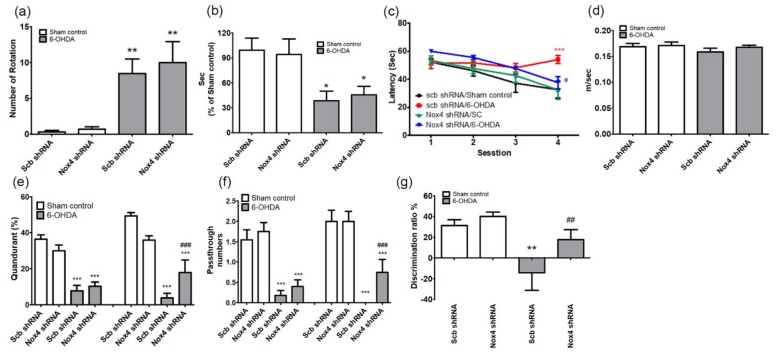
Inhibition of NOX4 reduces the memory impairment in the 6-OHDA-induced PD mouse. (**a**) Total apomorphine (APO)-induced rotation numbers; (**b**) retention times in rotarod tests were measured at 4 weeks after 6-OHDA injection in Scb shRNA/AAV2 or NOX4 shRNA/AAV2-injected mice. *n* = 12. * *p* < 0.05, ** *p* < 0.01 vs. sham control. (**c**–**f**) Spatial memory evaluation using the time latency (**c**), swimming speed (**d**), percentage of time in target quadrant (**e**), and (**f**) number of passes through the platform. (**g**) The novel object recognition test was performed, exploratory time spent for novel objects was recorded, and the discrimination ratio was calculated. *n* = 12 per group, * *p* < 0.05, ** *p* < 0.01, *** *p* < 0.001 vs. Scb shRNA sham control; # *p*< 005, ## *p*< 0.01, ### *p*< 0.001 vs. Scb shRNA 6-OHDA.

**Figure 8 ijms-20-00696-f008:**
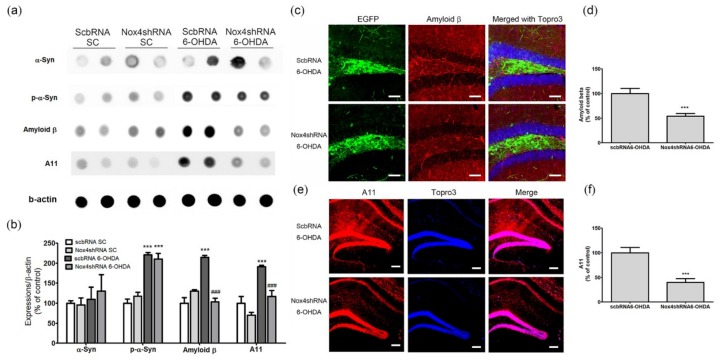
Inhibition of NOX4 reduced expressions of amyloid β and A11 oligomers in the hippocampus of 6-OHDA-injectecd PD mice. Representative immunoblot analyses of (**a**) α-syn, p-α-syn, amyloid β, A11 oligomers, and β-actin in the hippocampus of 6-OHDA-injectecd PD mice. (**b**) All expression levels were quantified using Quantity One software and normalized against β-actin. The results are expressed as a percentage of control. Data are shown as the mean ± SEM. *n* = 6 per group, *** *p* < 0.001, vs. Scb shRNA SC; ### *p* < 0.001 vs. Scb shRNA 6-OHDA. (**c**) Representative photomicrographs of cells expressing EGFP (green) represent AAV-transduced cells. Scb shRNA or NOX4 shRNA with amyloid β-positive cells is demonstrated by yellow staining after merging green (EGFP) and red (amyloid β) images. Blue colors show the Topro3 nuclear marker. (**d**) Yellow stained cells were counted and quantified using Quantity One software. The results are expressed as a percentage of control. Data are shown as the mean ± SEM. *n* = 6. (**e**) Representative photomicrographs A11 oligomers (red) staining in the hippocampal sections of Scb shRNA 6-OHDA and NOX4 shRNA 6-OHDA mice. Blue colors show the Topro3 nuclear marker. (**f**) The expression level was quantified using Quantity One software. The results are expressed as a percentage of control. Data are shown as the mean ± SEM. *n* = 6. *** *p* < 0.001 vs. Scb shRNA 6-OHDA. SC, sham control; Scb RNA, scrambled shRNA; scale bars = 50 μm.

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
