# Peer review of "The Role of NOX4 in Parkinson’s Disease with Dementia"

_ijms, 2019, doi:10.3390/ijms20030696_

Round 1
Reviewer 1 Report
Choi and coauthors have investigated the role of Nox4 in cognitive impairment, alpha-syn and Ab expression and aggregation in murine Parkinson disease model of dementia induced by injection of 6-OHDA. They found that nigrostriatal dopamine neurons had increased expression of alpha-syn and Nox1, while hippocampus had increased phosphorylation of a-syn, Abeta, oligomer A11 and Nox4 in mice injected with 6-OHDA. This was associated with cognitive impairment. The knockdown of Nox4 affects the Ab expression and A11 oligomer production and reduced the cognitive impairment.
This is a very good study on an important topic. In general, I found the manuscript well prepared and the experiments carefully designed and mapped out. Data are convincing and well presented and discussed. There some concerns regarding this study:
1. 6-OHDA mostly affects the mitochondrial pathways and oxidative metabolism. The arise question is whether the Nox1 and Nox4 is elevated due to specific effect of 6-OHDA or it is present in other models of PDD. The authors should address potential technical limitation of the model.
2. It will be very important to additionally analyse/calculate correlation between lesions, NOx1, Nox4, Abeta expression and A11 production and behavior changes.
Author Response
We are grateful to reviewer’s insightful and instructive comments and have made significant revisions accordingly. The most of points raised by reviewer were indeed valid and thus might be constructive to further strengthen our manuscript. Below is our point-by-point response to reviewer’s comments.
Comments and Suggestions for Authors
Choi and coauthors have investigated the role of Nox4 in cognitive impairment, alpha-syn and Ab expression and aggregation in murine Parkinson disease model of dementia induced by injection of 6-OHDA. They found that nigrostriatal dopamine neurons had increased expression of alpha-syn and Nox1, while hippocampus had increased phosphorylation of a-syn, Abeta, oligomer A11 and Nox4 in mice injected with 6-OHDA. This was associated with cognitive impairment. The knockdown of Nox4 affects the Ab expression and A11 oligomer production and reduced the cognitive impairment.
This is a very good study on an important topic. In general, I found the manuscript well prepared and the experiments carefully designed and mapped out. Data are convincing and well presented and discussed. There some concerns regarding this study:It has been well established that the brain infraction expands and reaches a peak about 1 week after photo-thrombotic (PT) induced-stroke before gradually declining thereafter. Please indicate why there is no recovery over the time course of 12 weeks after the PT in the current study.
1. 6-OHDA mostly affects the mitochondrial pathways and oxidative metabolism. The arise question is whether the Nox1 and Nox4 is elevated due to specific effect of 6-OHDA or it is present in other models of PDD. The authors should address potential technical limitation of the model.
Thank you for your advice. As your advice, the following sentences are included in the discussion section.
In addition, it is necessary to overcome the limitations of the model by conducting studies using PDD models other than the 6-OHDA administration model, but our study may be useful for preclinical studies of new diagnostic and therapeutic technologies for PDD.
2. It will be very important to additionally analyse/calculate correlation between lesions, NOx1, Nox4, Abeta expression and A11 production and behavior changes.
Thank you for your advice. As your advice, the following figure and sentences are included in the results section.
To investigate the effects of these molecules on cognitive deficits in PDD model with 6-OHDA, changes in NOX4, NOX1, Aβ expression and A11 oligomer production in hippocampus and behavioral changes of 6-OHDA injected mice were analyzed by Pearson correlation coefficient. There was a high correlation between cognitive impairment and A11 oligomer production, NOX4, Aβ expression except for NOX1 expression (Figure 5).
Figure 5. Relationship between the cognitive impairment and % changes of NOX4 (a), NOX1 (b), and Aβ expressions (c), and A11 oligomer production (d) in the hippocampus of 6-OHDA-injectecd PD mice.

Reviewer 2 Report
In this paper, Authors examined the role of NOX4 in Parkinson's disease with dementia.
This work is interesting and scientifically sound. The experimental design is globally well described and the conclusions are supported by the obtained results. However, there are some major concerns that Authors should address before the manuscript can be reconsidered for publication
In my opinion, the most important concern of this paper is related to the presentation of the Western Blotting results. More specifically:
-Fig. 1C: it is not clear for me why the backgrounds of the shown parts of the blots is so different from a blot to another. This should be explained by the Authors. Moreover, the molecular weights of the different considered proteins should be added in this figure. Also, it is not clear for me if this experiment also included the evaluation of NOX2 expression and, if yes, why Authors did not show the blot related to this NADPH oxidase isform;
-Fig. 4: the different molecular weights should be added in the figure. Again the background of the alpha-sin blot appears to me different from the others.
The other major concerns are related to the description of the immunohistochemical procedures. Indeed, as far as I can read, this section lacks from important detailes, such as the dilutions of the primary and secondary antibodies. Moreover, more informations about the microscope used by the Authors should be provided.
In the description of the Novel Object Recognition test, Authors should provide a reference for the 3h intervals.
In addition, for all performed ANOVAs, the obtained F values (and degrees of freedom) should be indicated (in the description of results or in the figure legends).
Author Response
We are grateful to reviewer’s insightful and instructive comments and have made significant revisions accordingly. The most of points raised by reviewer were indeed valid and thus might be constructive to further strengthen our manuscript. Below is our point-by-point response to reviewer’s comments.
Comments and Suggestions for Authors
In this paper, Authors examined the role of NOX4 in Parkinson's disease with dementia.
This work is interesting and scientifically sound. The experimental design is globally well described and the conclusions are supported by the obtained results. However, there are some major concerns that Authors should address before the manuscript can be reconsidered for publication
In my opinion, the most important concern of this paper is related to the presentation of the Western Blotting results. More specifically:
1. Fig. 1C: it is not clear for me why the backgrounds of the shown parts of the blots is so different from a blot to another. This should be explained by the Authors. Moreover, the molecular weights of the different considered proteins should be added in this figure. Also, it is not clear for me if this experiment also included the evaluation of NOX2 expression and, if yes, why Authors did not show the blot related to this NADPH oxidase isform;
2. Fig. 4: the different molecular weights should be added in the figure. Again the background of the alpha-sin blot appears to me different from the others.
Thank you for your advice. As your advice, there are corrected the result section.
Figure 1. Evaluations of motor performance, dopaminergic neuronal death, expressions of a-synuclein and NOX1 in the 6-OHDA-induced PD mouse
Figure 4. Expressions of p-a-syn, Amyloid b, A11 oligomers, and NOX4 in the hippocampus of 6-OHDA-injectecd PD mice.
3. The other major concerns are related to the description of the immunohistochemical procedures. Indeed, as far as I can read, this section lacks from important detailes, such as the dilutions of the primary and secondary antibodies. Moreover, more informations about the microscope used by the Authors should be provided.
Thank you for your advice. As your advice, there are corrected in the Method section.
4.4. Immunohistochemistry
Following perfusion with saline and 4 % paraformaldehyde in phosphate-buffered saline (PBS, GibcoBRL, Gaithersburg, MD, USA), brains were removed, and the forebrain and midbrain blocks were immersion-fixed in 4 % paraformaldehyde and cryoprotected in sucrose. Serial coronal sections (30 μm) were cut on a cryostat, collected in cryopreservative, and stored at -20 ºC. For immunolabeling studies, sections were incubated with a blocking solution (5 % horse serum and 0.3 % Triton X-100 in PBS, pH 7.5) and then with primary antibody at 4°C overnight. Mouse monoclonal anti-tyrosine hydroxylase (TH) antibody (MAB318, 1:2000) was obtained from EMD Millipore (Billerica, MA, USA). Next, sections were incubated with biotinylated goat anti-mouse IgG secondary antibodies (BA-9200,1:200, Vector Laboratories, Burlingame, CA, USA) in a blocking solution at room temperature for 1 h. The sections were incubated with avidin–biotin–peroxidase complex (Vector Laboratories, Burlingame, CA, USA) in PBS/Triton X-100 at room temperature for 1 h. Signal labeling was achieved with 0.05% 3,30-diaminobenzidine and 0.003% H2O2 (Vector Laboratories). Signals were evaluated on an inverted light microscope with 10x or 20x objectives (Carl Zeiss, Jena, Germany) [27].
4.7. Double-Fluorescence Immunostaining of Tissues
Free-floating sections (30 ÎĽm) were incubated in 0.1 M PBS containing 5% normal donkey serum and 0.3% Triton X-100 for 1 h, and subsequently incubated overnight with specific primary antibodies (Tuj-1, 1:1000, BioLegend, San Diego, CA, USA; a-syn, 1:1000, BD Bioscience, USA; p-a-syn, 1:1000, Abcam, Cambridge, UK, Amyloid beta, Ab, 1:1000, BioLegend (6E10), San Diego, CA, USA; A11, 1:1000, Invitrogen, Waltham, MA USA; NOX4 (1:500, Santa Cruz Biotechnology, USA) in 2% normal donkey serum (Vector Laboratories, Burlingame, CA, USA) in PBS at 4 and incubated with a 1:200 dilution of Alexa Fluor-conjugated donkey anti-rabbit (488) or donkey anti-mouse (546) antibodies (Invitrogen, Grand Island, NY, USA) for 1 h at room temperature and mounted on glass slides using Vectashield (Vector Laboratories, Burlingame, CA, USA). Fluorescent signals were evaluated on a confocal microscope (LSM 710, Carl Zeiss, Oberkochen, Germany) [1].
4. In the description of the Novel Object Recognition test, Authors should provide a reference for the 3h intervals.
Thank you for your advice. As your advice, there were added a reference for the 3h intervals in the method section.
4.12. Novel Object Recognition Test
The mice were placed in a 40x40x40 cm (height) sized box and allowed to equilibrate to the testing area 10 min per day for 3 days. After 24 h, two objects were placed in the testing area and then, the mouse was allowed to explore the two objects in the testing area for 5 min before being returned to the cage. After a 3 h-interval, one of the objects was either relocated or replaced with a new object and the mice were allowed to explore the testing area once again for 2 min [23,59].
5. In addition, for all performed ANOVAs, the obtained F values (and degrees of freedom) should be indicated (in the description of results or in the figure legends).
Thank you for your advice. As your advice, F values (and degrees of freedom) were indicated in the description of results.

Round 2
Reviewer 2 Report
The authors have addressed all my concerns.